# The critical role of point defects in improving the specific capacitance of δ-MnO$_2$ nanosheets

Peng Gao[1], Peter Metz[1], Trevyn Hey[1], Yuxuan Gong[1], Dawei Liu[1], Doreen D. Edwards[1], Jane Y. Howe[2], Rong Huang[3] & Scott T. Misture[1]

3D porous nanostructures built from 2D δ-MnO$_2$ nanosheets are an environmentally friendly and industrially scalable class of supercapacitor electrode material. While both the electrochemistry and defects of this material have been studied, the role of defects in improving the energy storage density of these materials has not been addressed. In this work, δ-MnO$_2$ nanosheet assemblies with 150 m$^2$ g$^{-1}$ specific surface area are prepared by exfoliation of crystalline K$_x$MnO$_2$ and subsequent reassembly. Equilibration at different pH introduces intentional Mn vacancies into the nanosheets, increasing pseudocapacitance to over 300 F g$^{-1}$, reducing charge transfer resistance as low as 3 Ω, and providing a 50% improvement in cycling stability. X-ray absorption spectroscopy and high-energy X-ray scattering demonstrate a correlation between the defect content and the improved electrochemical performance. The results show that Mn vacancies provide ion intercalation sites which concurrently improve specific capacitance, charge transfer resistance and cycling stability.

[1] Kazuo Inamori School of Engineering, Alfred University, Alfred, New York 14802, USA. [2] Hitachi High-Technologies Canada, Inc., 89 Galaxy Blvd, Suite 14, Toronto, Ontario, Canada M9W 6A4. [3] Cornell High Energy Synchrotron Source, Cornell University, Ithaca, New York 14853, USA. Correspondence and requests for materials should be addressed to S.T.M. (email: misture@alfred.edu).

Manganese dioxide ($MnO_2$) in its many forms has been the subject of much study for electrochemical capacitor applications[1,2]. In general, supercapacitors can be classified into two types: (i) electrical double-layer capacitors, which depend on charge separation at the electrode/electrolyte interface without Faradic process; and (ii) pseudocapacitors, which depend on Faradic redox reactions[3–5]. 'Extrinsic' pseudo-capacitance has recently emerged as a subclassification of materials that host ion intercalation but are engineered to short length scales to reduce diffusion distances such that the discharge behaviour becomes linear and no structural phase changes occur[6,7].

The birnessite form of $MnO_2$ ($\delta$-$MnO_2$), comprising stacked sheets of edge-shared $MnO_6$ octahedra with interlayer alkali ions[8–10], has been studied for some time and shows both double-layer and Faradaic charge storage[1,11,12]. Two-dimensional (2D) $\delta$-$MnO_2$ generally exhibits improved capacitance and rate behaviour compared to other polymorphs because the interlayer galleries provide high-speed pathways for diffusion of protons or alkali cations during the charge and discharge processes[13–15]. However, the low electrical conductivity of $\delta$-$MnO_2$ ($10^{-5}$ to $10^{-6}\,S\,cm^{-1}$) has greatly limited its application[16], prompting study of composite electrodes containing graphene[17,18], carbon nanotubes[19,20], carbon fibres[21,22] and so on. In addition, nanostructuring has been employed to improve the surface area and capacitance, for example by growing nanoneedles[23], nanoflowers[24], nanoparticles[25] and so on.

In recent years, intentional creation of cation vacancies has been explored to increase the charge storage capacitance of transition metal oxides, where cation vacancies provide additional cation intercalation sites[26]. Cation vacancy content may be controlled via aliovalent cation substitution[27], anion substitution[28], reducing or oxidizing heat treatments[29], or by equilibrating the oxides in pH-controlled suspensions[30]. The first studies correlating cation vacancies and charge storage properties were published by Ruetschi in the mid-1980s for intergrowth $\gamma$-$MnO_2$ phases[31–33]. Metal vacancy content can be quite large in some cases, for example, Wei et al.[28] modified anatase $TiO_2$ with monovalent $F^-$ and $OH^-$ anions to form up to 22 at% $V_{Ti}''''$ for additional Li storage. Similarly, Koo et al.[34,35] transformed $Fe_3O_4$ (spinel) into hollow $\gamma$-$Fe_2O_3$ rods containing up to 44% vacant iron sites, and showed that Li and Na ion intercalation is possible without structural phase transformations.

Little work on the formation of cation vacancies in $\delta$-$MnO_2$ nanosheets has been reported to date, but extensive study of the important role of birnessite-like $MnO_2$ in photosynthesis, ion sorption and other bio- and geochemical processes provides rich literature on its behaviour in aqueous environments[36]. Most recently, Manceau et al.[30], building from many earlier works on Mn oxidation state in K-birnessites[10,37], systematically investigated the effects of pH on cation vacancy content in phyllomanganate nanoparticles. Their approach exploited high-energy X-ray scattering to perform pair distribution function (PDF) analysis, truncation rod analysis and simulations using the Debye equation to show that lower pH causes migration of Mn from the $MnO_2$ nanosheet to the sheet surface. Thus, the authors were able to quantify Mn vacancies in the nanosheets and $Mn^{3+}$ cations displaced to the interlayers. Later work by Marafatto and coworkers[36] employed sub-picosecond optical and X-ray absorption spectroscopy to track the mechanisms of Mn reduction under illumination to quantify the effects of different interlayer cations on $MnO_2$ photo-reduction rates. Their results support the Mn redox reaction mechanism proposed by Manceau et al.[30], including displacement of the $Mn^{3+}$ cation to the interlayer gallery.

Additional recent work has been focused on determining mechanisms of charge storage. For example, in situ X-ray absorption studies have been used to show when the Faradaic reactions occur in $MnO_2$ nanosheets[38] and to track the average Mn oxidation state across wide voltage swings[39–41]. Similarly, in situ Raman spectroscopy has been applied to track changes in the vibrational bands and has highlighted cation size effects over the Li, Na and K series[42].

In the present work, we exfoliate and reassemble $\delta$-$MnO_2$ nanosheets to form 3D macroporous pseudocapacitive electrodes with controlled concentration of Mn point defects and $Mn^{3+/4+}$ ratios. Electrochemical and high-energy X-ray scattering measurements provide direct evidence that intentional Mn ion defects and Mn reduction synergistically improve supercapacitor performance. The three-dimensional (3D) assembly and defect control represent straightforward and industrially scalable approaches to improving specific capacitance.

## Results

**Phases and microstructure.** The morphologies of pristine $K_xMnO_2$, and its protonated form, $H_xMnO_2$, are shown in Supplementary Fig. 3. From Supplementary Fig. 3a,b it can be seen that the $K_xMnO_2$ and $H_xMnO_2$ particles are platy, with lateral dimensions in the range of one micron, and that proton exchange does not alter the grain morphology. High-magnification SEM and TEM images are shown in Fig. 1a–c for an individual nanosheet (Fig. 1a,b) and the reassembled $MnO_2$ nanostructure (Fig. 1c). The images demonstrate that the sheets are generally flat, with some scrolling at the edges as has been noted in other nanosheet studies[43,44]. The atomic force microscopy (AFM) image in Supplementary Fig. 4 also displays nanosheet fragments with flat surfaces and thickness of five nanosheets, suggesting that some of the nanosheets exfoliate into bunches or restack to a small extent upon drying. The flocculated nanosheet samples exhibit 3D porous nanostructures (Fig. 1d; Supplementary Fig. 3c) and equilibration at different pH values have no influence on their morphologies. The observation of porous structures after reassembly highlights the usefulness of our ultrasonic-assisted exfoliation and flocculation procedure, where the exfoliation rate is greatly enhanced compared with other reported procedures[45,46].

Synchrotron diffraction data for the parent $K_xMnO_2$ is shown in Supplementary Fig. 5. Rietveld refinement reveals that the as-synthesized parent material is layered birnessite, primarily exhibiting the monoclinic polytype with less than 10 wt% of the rhombohedral polytype and no additional crystalline phases. The X-ray diffraction patterns (Fig. 2) for protonated and reassembled $MnO_2$ demonstrate that phase purity was achieved during synthesis and that exfoliation and reassembly yields complex nanostructures as evidenced by peak broadening and asymmetry. After reassembly of the nanosheets, in-plane $hk0$ reflections remain discernible, indicating that the crystalline nature of the 2D sheets is preserved. Further, the derived PDF is consistent both with literature[47] and the calculated PDF of a single $\delta$-$MnO_2$ nanosheet (Supplementary Fig. 6), further confirming the $\delta$-$MnO_2$ nanosheet motif is maintained. The broadening of the $00l$ basal reflections shows that although some sheet-to-sheet restacking occurs, the stacks are on the order of only 3–4 nm (estimated using the Scherrer equation), which is consistent with the AFM results. Shifts of the basal reflections result from increased water content in the reassembled nanosheet floccules compared with the proton-exchanged form.

Nitrogen adsorption–desorption isotherms were used to quantify the specific surface area (SSA) of all specimens as shown in Fig. 3. While protonated $MnO_2$ showed no evidence of

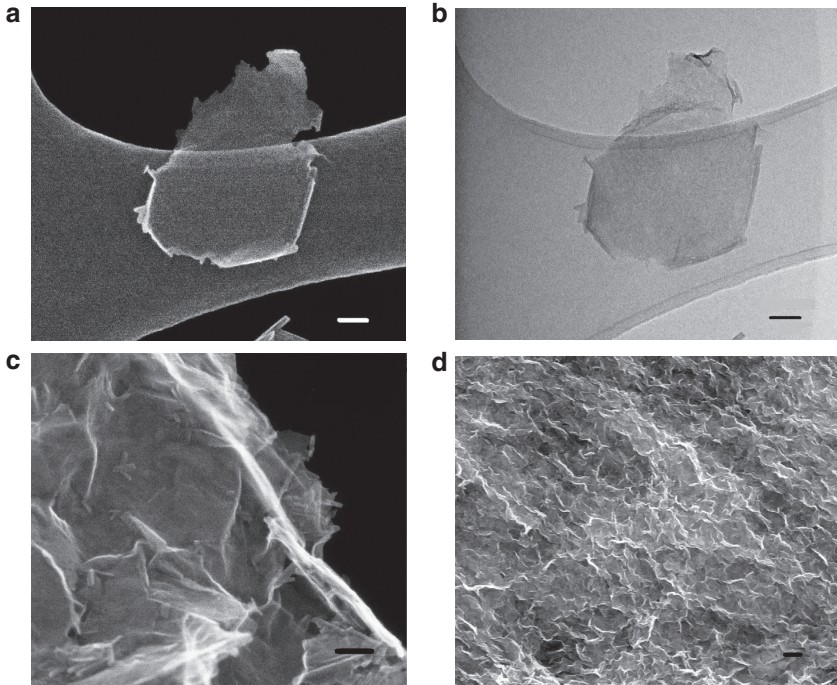

**Figure 1 | Electron microscopy of δ-MnO₂ nanosheets.** (**a**) SEM image and (**b**) bright-field TEM image of exfoliated MnO₂ nanosheets, (**c**) high-magnification SEM image and (**d**) SEM image of reassembled MnO₂ nanostructures treated in pH = 2 solution for 24 h. Scale bar, 50 nm (**a–c**); 500 nm (**d**).

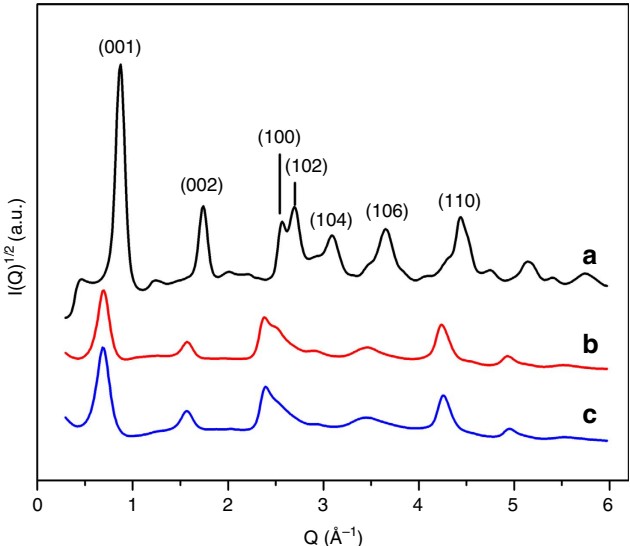

**Figure 2 | Powder diffraction patterns of protonated and reassembled δ-MnO₂.** XRD patterns of (**a**) protonated MnO₂, (**b**) reassembled MnO₂ treated in pH = 4 solution for 24 h, (**c**) reassembled MnO₂ treated in pH = 2 solution for 24 h. Data collected on APS 11-ID-B.

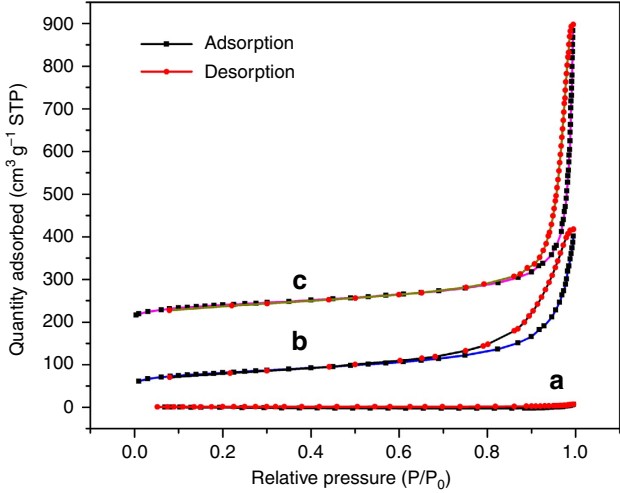

**Figure 3 | N₂ adsorption isotherms of protonated and reassembled δ-MnO₂.** N₂ adsorption–desorption isotherms of (**a**) protonated MnO₂, (**b**) reassembled MnO₂ treated in pH = 2 solution for 24 h, (**c**) reassembled MnO₂ treated in pH = 4 solution for 24 h. Curves (**b**) and (**c**) are offset by 50 and 200 cm³ g⁻¹ STP, respectively.

mesopores, the reassembled MnO₂ nanostructures show typical type IV isotherms (IUPAC classification) with distinct H3-type hysteresis loops, a result of open slit-like mesopores[48]. The Brunauer Emmet and Teller (BET) SSA were $120 \pm 0.4 \, \mathrm{m^2 \, g^{-1}}$ for the pH = 2 sample and $144 \pm 1 \, \mathrm{m^2 \, g^{-1}}$ for the pH = 4 sample, and only $4.5 \pm 0.1 \, \mathrm{m^2 \, g^{-1}}$ for $H_xMnO_2$, where the former values are roughly double the surface areas reported by Song *et al.*[49] for reassembled oxide nanosheets. Altogether, the XRD, BET and microscopy show that the reassembled nanosheets have macro- and mesopores, with the mesoporosity arising due to loose agglomeration of randomly oriented sheet clusters. The extent of

exfoliation and/or restacking, typical sheet thicknesses determined by AFM and average crystallinity in the sheet stacking direction as determined by XRD are similar or better for our MnO₂ specimens than those reported recently for MnO₂, TiO₂, Co₃O₄, ZnO and WO₃, for example[44]. Therefore, the typical structures in Fig. 1d are of the form of edge-to-face assembled nanosheet booklets, with 'wall thicknesses' of up to 4 nm. The high SSA of the MnO₂ nanosheet assemblies facilitates infiltration of the electrolyte into the porous electrode to enhance the specific capacitance[6,50].

During soft chemical processing, the surface and interlayer tetrabutyl ammonium hydroxide (TBAOH) and water content

may vary with processing conditions. The water content in protonated, $pH = 2$ and $pH = 4$ samples was obtained by thermogravimetric analysis (Supplementary Fig. 7). The presence of structural water enables rapid proton or alkali cation transport within the interlayer, which is beneficial for increasing the charge storage properties[6]. The slightly higher water content in the $pH = 4$ versus $pH = 2$ sample is a result of its slightly larger surface area. The presence of residual TBAOH in the reassembled samples was indicated by a small mass loss ($\sim 3\%$, corresponding to 0.01 TBAOH per $MnO_2$ formula unit and therefore about 9% surface coverage) in the temperature region of 200–400 °C (refs 51,52). Infrared spectroscopy (Supplementary Fig. 8) and X-ray photoelectron spectroscopy (XPS) (Supplementary Fig. 9) were also used to detect remnant TBAOH in the nanosheet assemblies after processing. Infrared was used to show that the TBAOH is removed from the nanosheet assemblies after one charge–discharge cycle, as shown in Supplementary Fig. 10. Although it is not possible to completely remove the TBAOH molecules from the nanosheet assemblies by washing, its small surface coverage ($\sim 9\%$) and easy extraction during the first electrochemical cycle together suggest that it has limited if any influence on the measured electrochemical performance.

**Redox and defects**. The oxidation state of the Mn ions in all samples was investigated using Mn K-edge X-ray absorption near-edge spectroscopy (XANES) and XPS. The XPS analysis is described in Supplementary Information (Supplementary Fig. 11). Figure 4a shows the edge spectra of standard materials including MnO, $Mn_3O_4$, $Mn_2O_3$ and $MnO_2$.

The XANES spectra of the $H_xMnO_2$, $pH = 2$ and 4 nanosheet assemblies, and reference materials $Mn_2O_3$ and $MnO_2$ are shown in Fig. 4b. The line profiles are characterized by features that correspond to a pre-edge range with two weak broad peaks at 6,540–6,545 eV, a main-edge range that has one inflection point A, and the resonance peak range B[49,53]. The weak pre-edge peaks P and P′ correspond to the dipole-forbidden $1s \rightarrow 3d$ transition[54]. All three samples exhibit higher intensity of peak P′ than for peak P, but the peak P′ is less intense as compared with the β-$MnO_2$ reference. This observation confirms the existence of mixed oxidation states of $Mn^{3+}/Mn^{4+}$ in the samples. Also, the higher intensity ratio of P′ to P for $H_xMnO_2$ compared with the $pH = 2$ and 4 samples indicates that it has less $Mn^{3+}$, because the relative intensity of the peaks P′/P is proportional to the average oxidation state[54].

The main absorption range can be assigned to the dipole-allowed $1s \rightarrow 4p$ transition. The associated edge energy is usually taken as the energy of the peak in the first derivative, which corresponds to the inflection point of the main edge in the XANES spectra (Fig. 4c). Clearly, the main absorption edge (A) progressively shifts to lower energies with decreasing pH, implying lower pH progressively reduces Mn to the trivalent state. Also, the presence of the intense peak B for the nanosheet assemblies indicates that they are mainly comprised of edge-shared $MnO_6$ octahedra[55]. This observation further confirms that the δ-$MnO_2$ lattice is not dissolved by equilibration in HCl at pH as low as 1.

The average oxidation state (AOS) of Mn was determined by establishing a linear relationship between the K-edge energy and Mn oxidation state (Fig. 4d). The AOS of Mn is 3.59 for $H_xMnO_2$,

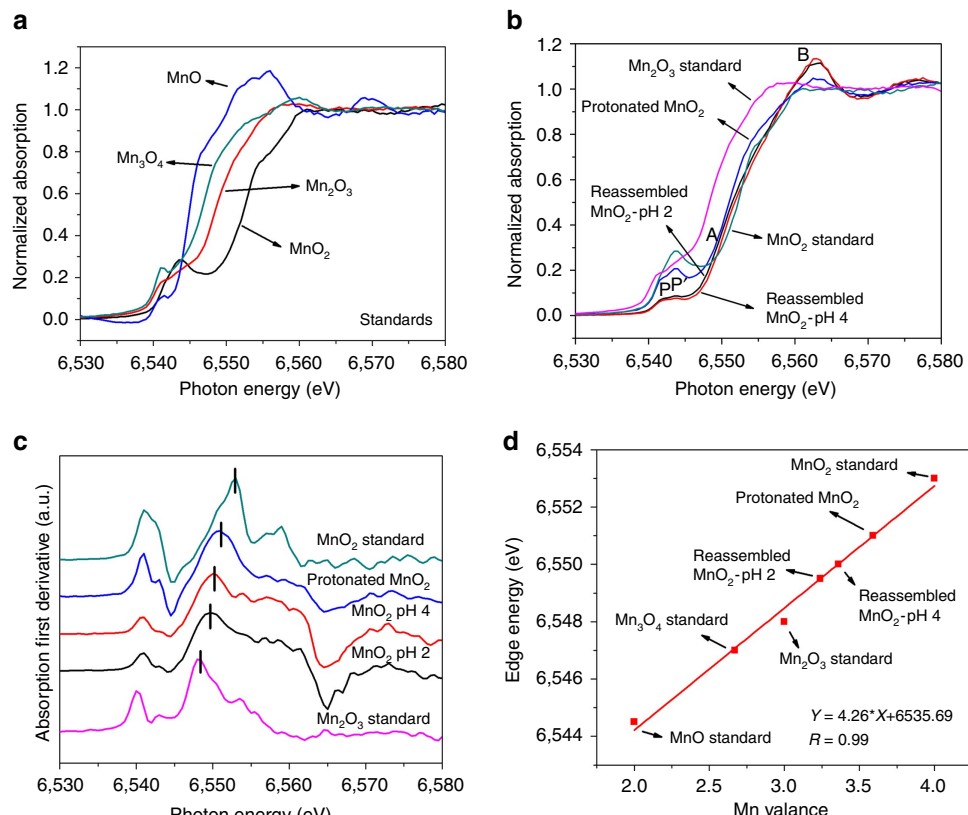

**Figure 4 | X-ray absorption spectra of $MnO_x$ standards and experimental specimens.** X-ray absorption measurements of as-prepared three samples and reference Mn oxide materials. (**a**) XANES spectra of reference materials MnO, $Mn_3O_4$, $Mn_2O_3$ and $MnO_2$. (**b**) XANES spectra of protonated $MnO_2$, $pH = 2$ and 4 treated reassembled $MnO_2$. The reference materials of $Mn_2O_3$ and $MnO_2$ from (**a**) are also shown for of ease comparison. (**c**) First derivative curves corresponding to the samples shown in **b**. (**d**) Average oxidation state of Mn for the samples and standards derived from the K-edge energy.

**Table 1 | Summary findings of defects and electrochemistry in δ-MnO₂ electrodes.**

| Samples | AOS | [Mn³⁺] (%) | [Mn⁴⁺] (%) | [V$_{Mn}$] (%) | S.S.A. (m² g⁻¹) | C$_p$ (F g⁻¹) | R$_{ct}$ (Ω) | Na : Mn (fract.) |
|---------|-----|-----------|-----------|---------------|-----------------|---------------|--------------|------------------|
| Reassembled MnO₂ - pH 2 | 3.24 | 76 | 24 | 26.5 | 120 | 306 | 3 | 0.28 |
| Reassembled MnO₂ - pH 4 | 3.36 | 64 | 36 | 19.9 | 144 | 209 | 15 | 0.19 |
| Protonated MnO₂ | 3.59 | 41 | 59 | 18.3 | 4.5 | 103 | 23 | 0.09 |

Comparison of defect structures and electrochemical supercapacitor properties of bulk protonated H$_x$MnO₂ and MnO₂ nanosheet assemblies.

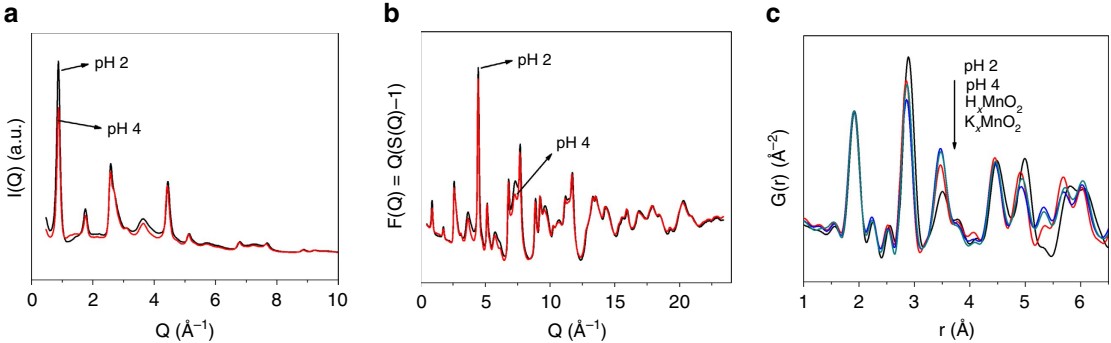

**Figure 5 | High-energy X-ray scattering and experimental PDF data.** (**a**) I(Q) and (**b**) F(Q) for the pH = 2 and pH = 4 treated MnO₂ nanosheets assemblies, and (**c**) G(r) for the pH-treated samples, protonated MnO₂ (H$_x$MnO₂), and parent phases (K$_x$MnO₂). The basal reflections evident in I(Q < 2 Å⁻¹) indicate a measure of restacking. The variation in intensity as a function of pH indicates a variation in the interlayer structure attributable to either surface water or Mn. The inverse trends in the Mn–Mn peak (2.89 Å) and Mn–Mn$^{IL}$ peak (3.45 Å) as a function of pH support the Mn³⁺ displacement model recommended by Manceau et al.[30]

3.36 for the pH = 4 MnO₂ nanosheet assembly, and 3.24 for the pH = 2 variant. These values are also listed in Table 1 for correlation with the PDF and electrochemical results discussed below. The dependence of Mn valence on pH generally follows that reported earlier by Manceau and coworkers[30] for bulk Na-saturated δ-MnO₂ (birnessite) powders, where the Mn AOS varied from 3.81 at pH = 9 to 3.69 for pH = 3. For the alkali-free MnO₂ samples studied herein, we find that treating the exfoliated MnO₂ assemblies at lower pH increases the Mn reduction to a greater extent than found for crystalline Na$_x$MnO₂ by Manceau and coworkers[30]. Furthermore, we find competing steric and thermodynamic effects by comparison of the proton-exchanged H$_x$MnO₂ to the exfoliated and reassembled variants. The AOS of H$_x$MnO₂ treated at pH < 1 is 3.59, whereas the exfoliated samples treated at higher pH have AOS values of 3.36 (pH = 4) and 3.24 (pH = 2). We attribute the lesser extent of Mn reduction in crystalline H$_x$MnO₂ to steric hindrance by the interlayer galleries which are crowded with protons and water. This has the two-fold effect of hindering access of aqueous H₃O⁺ to reducible Mn⁴⁺, and of preventing displacement of Mn³⁺ to the sheet surface. Indeed, earlier work by Gaillot et al.[10] noted that during thermal reduction of Mn⁴⁺ $\xrightarrow{350\,°C}$ Mn³⁺ in crystalline K-birnessite, Mn vacancies were not formed despite the unfavourable in-sheet lattice strain due to Jahn–Teller distortions inherent to Mn³⁺. The interactions of neighbouring in-plane Mn³⁺ and Mn⁴⁺ sites therefore contributes both strain and electrostatic-driven components to the energetics of δ-MnO₂ defect equilibration.

High-energy X-ray scattering and pair distribution function analysis were undertaken to further characterize the defects formed in soft chemically reduced MnO₂ nanosheets, with the specific goal of correlating the quantity of Mn point defects with Mn reduction and electrochemical performance. PDF analysis probes not only the local atomic bonding motifs but also intermediate and long-range order, thus making it an appropriate tool for investigating the structures of poorly-crystalline materials

and nanoparticles that yield diffraction patterns with large amounts of diffuse scattering[47].

The observed scattered intensity, the reduced structure function F(Q) and the associated PDF for the reassembled nanosheets are shown in Fig. 5. We focus our attention on two PDF correlation peaks: the in-plane Mn–Mn peak at 2.89 Å and the Mn–Mn$^{ooo}_{surf.}$ distance at 3.45 Å (Fig. 6a). Two notable advantages of the PDF method are that these two Mn correlations are unique in the alkali-free δ-MnO₂ structure (Fig. 6b) and that the integrated area of a PDF peak represents the number density of that specific correlation type[56]. We leverage these facts to obtain useful estimates of Mn vacancy concentrations in lieu of developing complete structure models, which are notably difficult for these material systems[57,58].

To compare the PDF with respect to different samples, we have normalized our data to the amplitude of the 1.9 Å Mn–O peak. This choice is based on the implicit assumption that all Mn ions are coordinated in a six-fold environment. We further assume that the intrinsic V$'''_{Mn}$ population in the parent phase K$_x$MnO₂ is on the order of parts per million, which will be negligible compared to the per cent-level vacancies in the MnO₂ nanostructures. Consequently, the relative in-plane Mn concentration can be calculated according to the following equation:

$$\frac{[Mn]_i}{[Mn]^0} = \frac{A_i}{A^0} \text{ or } [Mn]_i = \frac{A_i}{A^0}. \tag{1}$$

Where the superscript $^0$ denotes quantities in K$_x$MnO₂, the subscript $_i$ denotes quantities for a derivative sample, and [Mn]$^0$ is taken as 1. The absolute accuracy of the values obtained using equation (1) will be influenced by the correlations between interlayer water or hydroxyl pairs, which perturb the low-r PDF region, leading to overestimation of the Mn$^{ooo}_{surf}$ content[47]. While there is notable difficulty in absolute quantification of Mn defects in these materials, this method should provide reasonable relative quantification in the derived nanosheet assemblies.

Although estimation of error is not possible with this empirical approach, a per cent level of uncertainty is expected for the calculated Mn concentrations.

As shown in Fig. 6a, the $V_{Mn}'''$ concentration, which is equivalent to the $Mn_{surf.}^{\circ\circ\circ}$ concentration, is then is $1 - [Mn]_i$ in fractional units. Here we define this type of defect as a 'surface Frenkel' defect, where displacement of the in-plane Mn to the nanosheet surface is reminiscent of the Frenkel defect vacancy-interstitial pair. The results of our analysis are summarized in Table 2. Consistent with the model of Manceau et al.[30], we observe an increase in $\left[V_{Mn}'''\right]$ with decreasing pH, as well as the appearance of a PDF peak at a distance not found in the δ-MnO$_2$ structure, which corresponds to $Mn - Mn_{surf.}^{\circ\circ\circ}$. The increase in concentration of surface Frenkel defects by ∼30% between the pH = 4 and pH = 2 samples supports the hypothesis that increased proton sorption at the MnO$_2$ surface in more acidic electrolytes expels more in-plane Mn leading to the formation of more Mn vacancies.

As noted for the Mn valence, the PDF analysis likewise shows an apparent contradiction in the surface Frenkel defect content of the crystalline H$_x$MnO$_2$ when compared to the defect content of the pH = 2 and 4 samples. As shown in Table 1, while the protonated form H$_x$MnO$_2$ is equilibrated at pH < 1, its defect concentration (18.3%) is smaller than that in the pH = 2 sample (26.5%) and is even smaller than that of the pH = 4 sample (19.9%). As noted earlier, previous work shows that Mn$^{3+}$, with its Jahn–Teller distortion, may be accommodated in MnO$_2$ sheets in crystalline birnessites[59]. Table 1 shows that ∼2/3, 1/2 and 1/4 of the Mn ions remaining in the nanosheets are reduced to Mn$^{3+}$ for the pH = 2, 4 and H$_x$MnO$_2$ samples, respectively. The PDF and XANES analyses are therefore complementary in demonstrating that surface Frenkel defects and Mn reduction are more favourable in high-surface area nanosheet assemblies, with steric limitations reducing the extent of the reactions in well-crystalline birnessites.

**Electrochemical measurements.** As shown in Fig. 7a, the cyclic voltammetry (CV) curves for all three samples exhibit largely rectangular shapes in the potential window from 0 to 1 V, which indicates capacitive behaviour. The absence of clear redox peaks for protonated and pH = 4 samples implies that the electrodes are charged and discharged at a pseudo-constant rate over the whole voltammetric cycle. However, the presence of broad redox peaks for the pH = 2 sample indicates that ion intercalation is an active and detectable charge storage mechanism for this specimen. Indeed, even broader redox peaks are noted for the pH = 3 sample (Supplementary Fig. 12a), while no redox peaks are found for the pH = 4 and pH = 9 (Supplementary Fig. 12b) samples. The marked increase in Mn surface Frenkel defects with decreasing pH appears to enhance the intercalation reaction. This is also reflected in Fig. 7b–d, which shows the galvanostatic charge–discharge curves for the samples between 0 and 1 V under different current densities. The potential does not linearly change with time, a behaviour typical for pseudocapacitive materials that involve redox reactions.

The specific capacitance of the electrode measured by the galvanostatic discharge method can be calculated as:

$$C = \frac{I\Delta t}{\Delta V m} \qquad (2)$$

where $C$ (F g$^{-1}$) is the specific capacitance, $I$ (A) is the constant discharge current, $\Delta t$ (s) is the discharge time, $\Delta V$ (V) is the potential window, and $m$ (g) is the mass of the active material in the electrode. The specific capacitances of the three samples obtained at different current densities are summarized in Fig. 7e. At 0.2 A g$^{-1}$, the specific capacitance is 306 F g$^{-1}$ for the pH = 2 sample, 209 F g$^{-1}$ for the pH = 4 sample and 103 F g$^{-1}$ for crystalline H$_x$MnO$_2$. With increasing current density, the specific capacitance decreases gradually for all the three samples.

The defect chemistry also has an impact on cycling stability, demonstrated using the pH = 2 and 4 nanosheet assemblies, which is shown in Fig. 7f. After 1,000 charge–discharge cycles using a very high-current density of 5 A g$^{-1}$ (2.5 mA cm$^{-2}$), the pH = 2 sample shows good stability, retaining 83% of its initial specific capacitance. In contrast, the pH = 4 sample retains only

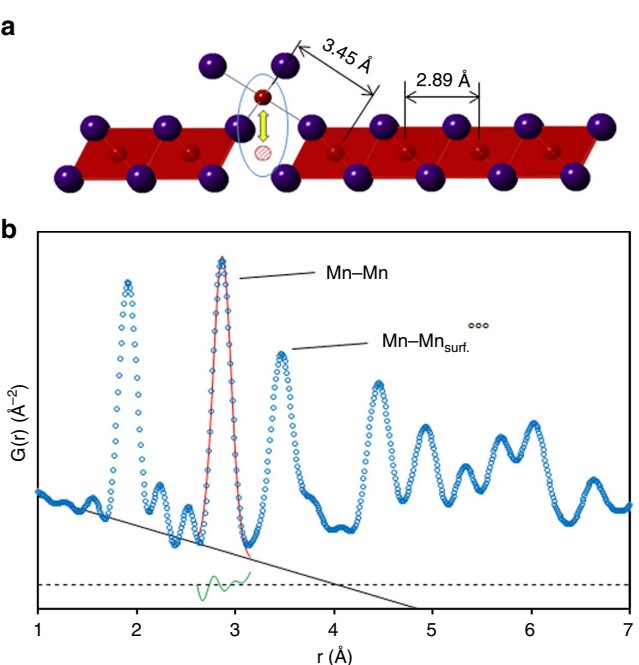

**Figure 6 | Empirical analysis of δ-MnO$_2$ PDF.** Using the resolved in-plane Mn-Mn correlation peak (**a**) we can empirically estimate the Mn surface Frenkel (circled) concentration using a Gaussian peak and linear baseline (**b**). The PDF amplitude is normalized to the Mn-O correlation peak; therefore the ratio of nanosheet Mn-Mn to K$_x$MnO$_2$ Mn-Mn peak areas represents the fractional Mn-occupancy of the nanosheet assembly. δ-MnO$_2$ equilibrated at pH = 2 is shown here.

**Table 2 | Estimation of surface Frenkel defect concentration.**

| Data set | Centre (Å) | Height (arb.) | Area (arb.) | FWHM (Å) | Mn-vac. (%) |
|---|---|---|---|---|---|
| Pristine MnO$_2$ | 2.897 | 15.988 | 3.461 | 0.203 | N/A |
| Protonated MnO$_2$ | 2.858 | 13.896 | 2.826 | 0.191 | 18.3 |
| Reassembled MnO$_2$–pH 4 | 2.876 | 13.810 | 2.772 | 0.189 | 19.9 |
| Reassembled MnO$_2$–pH 2 | 2.867 | 12.335 | 2.544 | 0.194 | 26.5 |

Gaussian parameters for the Mn–Mn in-sheet PDF peak corresponding to different samples extracted using Fityk[70]. Peak area is normalized to the Mn vacancy content of the parent phase K$_x$MnO$_2$ with the assumption that the intrinsic $\left[V_{Mn}'''\right]$ population is on the order of parts per million in the high-temperature crystalline phase.

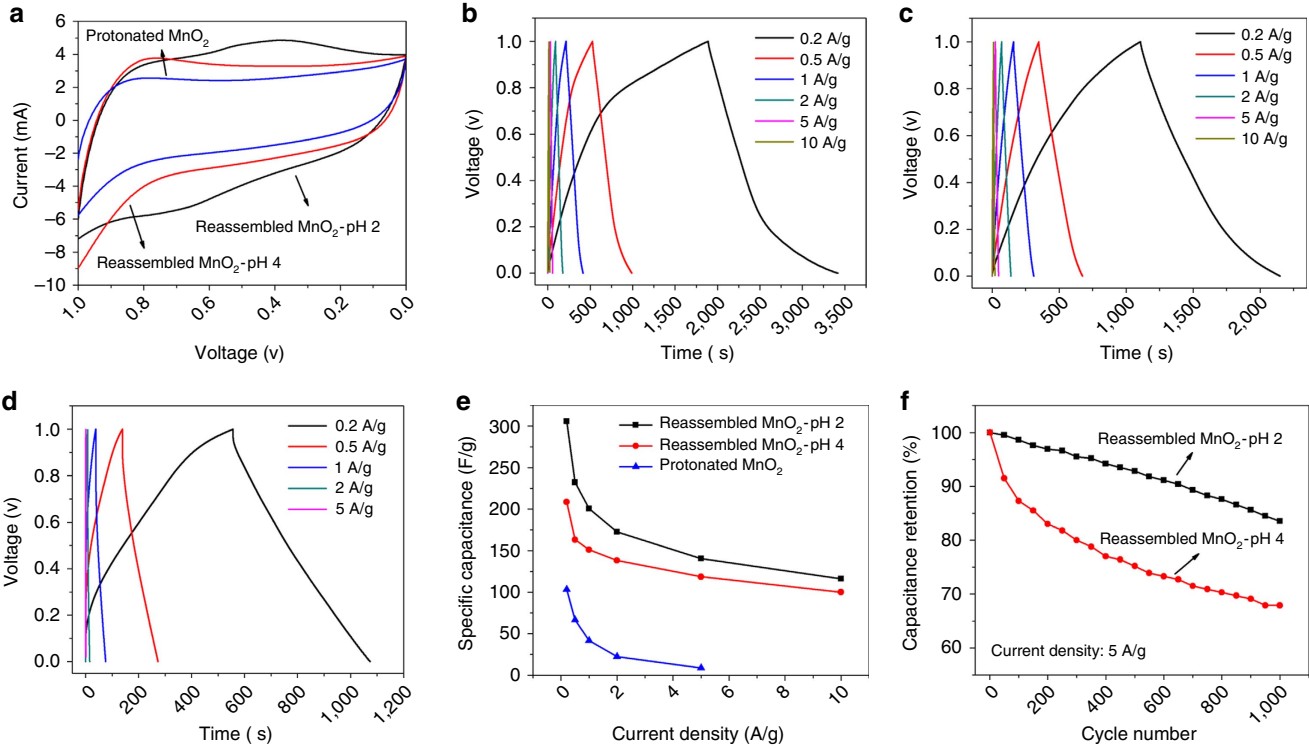

**Figure 7 | Electrochemical performance of reassembled δ-MnO₂ electrodes.** Electrochemical testing results: (**a**) CV curves for the samples at 50 mV s⁻¹ scan rate; (**b**), (**c**) and (**d**) galvanostatic charge–discharge curves of the pH = 2, 4 and HₓMnO₂ samples; (**e**) comparison of specific capacitance for the samples as a function of current density; and (**f**) cycle stability at constant current density of 5 A g⁻¹ between 0 and 1V.

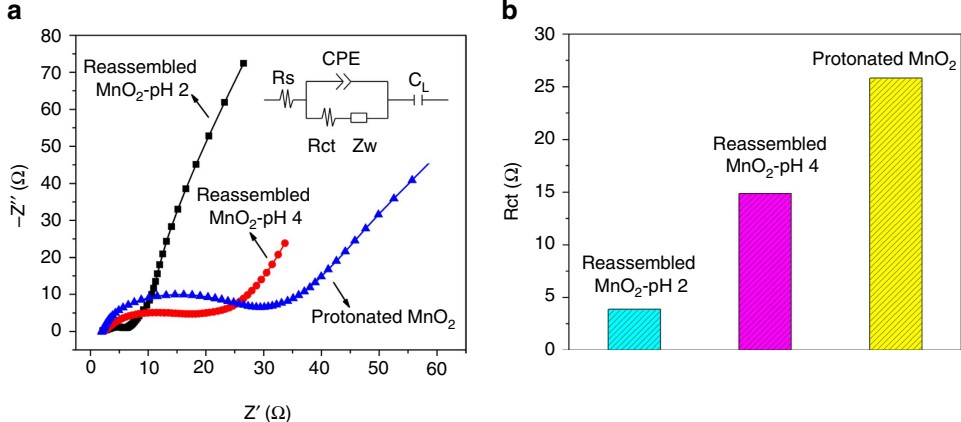

**Figure 8 | Electrochemical impedance spectroscopy of reassembled δ-MnO₂ electrodes.** (**a**) Nyquist plots of the samples in the frequency range of 0.1 Hz–100 kHz at an open circuit potential of 5 mV (inset shows the electrical equivalent circuit used for fitting the impedance spectra); and (**b**) comparison of the charge transfer resistance among the three samples (values obtained from the fitted data).

68% of its initial capacitance. Our ongoing work involves tracking the reversibility of the electrochemical strains induced during cycling, and although we have not yet uncovered the origins of the cycling fade, our results are promising when compared with earlier work, for example for MnO₂ nanostructures which retain ∼92% of the initial capacity after 1,000 cycles when using a current density five times smaller[49].

Electrochemical impedance spectroscopy (EIS) was employed to measure the charge transfer resistance of each electrode with the results shown in Fig. 8a. All plots exhibit a straight line in the low-frequency region and a single semicircle in the high-frequency region, indicating a diffusion-limited step in the

low frequency region and a charge transfer limited step in the high-frequency region[60]. The Nyquist plots were further modelled and interpreted by using an appropriate electrical equivalent circuit, which is shown in the inset in Fig. 8a. $R_s$ is a combined resistance including ionic resistance of the electrolyte, intrinsic resistance of the substrate and contact resistance at the active material/current collector interface. $R_{ct}$ is the charge transfer resistance caused by the Faradaic reaction. $Z_w$ is the Warburg resistance which is related to ion diffusion in the electrolyte, $CPE$ is a constant phase element and $C_L$ is the limit capacitance[61–63]. The calculated charge transfer resistances ($R_{ct}$) extracted from the high-frequency range were 23, 15 and 3 Ω for

the $H_xMnO_2$, pH = 4 and pH = 2 samples, respectively, as shown in Fig. 8b.

Table 1 summarizes the characteristics of the electrodes and their properties, showing several clear trends. First, the XPS and XANES studies, combined with quantification of surface Frenkel defects, demonstrate that the $Mn^{3+/4+}$ ratio in the nanosheets trends with pH treatment. More specifically, $Mn^{4+} \rightarrow Mn^{3+}$ reduction does not only result in formation of surface Frenkel defects but some of the Mn within the sheets is reduced to the trivalent state. As calculated based on the data in Table 1, ∼2/3, 1/2 and 1/4 of the Mn ions remaining in the nanosheets are reduced to $Mn^{3+}$ for the pH = 2, 4 and $H_xMnO_2$ samples, respectively.

The specific capacitance for the reassembled nanosheets correlates with the surface Frenkel defect population and $Mn^{3+}$ content, as well as charge transfer resistance. For example, the surface areas of the pH = 2 and 4 samples are similar, but the sample treated at pH = 2 has 50% higher capacitance, 47% more $Na^+$ ion intercalation, ×5 smaller charge transfer resistance, with 33% more surface Frenkel defects, highlighting the importance of the cation defects on $Na^+$ ion intercalation.

The surface Frenkel defect provides two likely intercalation sites: the Mn ion vacancy, which is accessible from one side of the nanosheet; and the undercoordinated $MnO_6$ surface octahedron on the opposite side of the nanosheet (Fig. 6a). In parallel, the defect reaction provides a large concentration of $Mn^{3+}$ cations, which can presumably participate in polaron hopping conduction, thus improving electrical conductivity and charge transfer efficiency[64,65]. The specific mechanism of $Na^+$ ion intercalation into the nanosheets is a topic of current study in our group, but we speculate that there exists some synergistic effect of the defect content and Mn redox that together define the charge transfer resistance and specific capacitance. The electrochemical cycling behaviours suggest that is likely that steric effects such as local distortions in the nanosheets around vacancies are active in relieving electrochemical strains that occur during cycling, and thus define to a large extent the specific capacitance.

## Discussion

Electrostatic assembly of δ-$MnO_2$ nanosheets in suspension under carefully controlled experimental conditions yields self-assembled 3D porous nanostructures with surface areas of ∼150 $m^2 g^{-1}$. Equilibrating the reassembled nanosheets in varied pH controls the extent of $Mn^{4+} \rightarrow Mn^{3+}$ reduction, as well as creating charged defect pairs we term 'surface Frenkel defects' comprising a Mn vacancy within the sheet and a six-fold coordinated $Mn^{3+}$ site on the surface of the nanosheet. The Mn surface Frenkel defect content reaches 26.5% for the nanosheet assemblies equilibrated at pH = 2 and 19.9% for the pH = 4 sample. The XPS and XANES data indicate an increase of the $Mn^{3+}/Mn^{4+}$ ratio with decreasing pH and the electrochemical results show direct correlation of Mn cation defects with specific capacitance. The specific capacitance increased from about 200 $F g^{-1}$ (pH = 4) to over 300 $F g^{-1}$ (pH = 2) by intentional introduction of ∼30% surface Frenkel defects, while at the same time the charge transfer resistance decreased from ∼15 Ω to ∼3 Ω. Therefore, it is now clear that Mn surface Frenkel defects in δ-$MnO_2$ nanosheets increase $Na^+$ ion intercalation by providing new, low energy intercalation sites.

## Methods

**Chemicals and reagents.** $MnCO_3$, $K_2CO_3$, $Na_2SO_4$, acetylene black, poly (vinylidene fluoride) (PVDF) and nickel foil were purchased from Alfa Aesar. 6N hydrochloric acid (HCl) solution and sodium hydroxide (NaOH) were obtained from Fisher Scientific. The tetrabutylammonium hydroxide solution (TBAOH,

40 wt% in $H_2O$) and N-methyl-2-pyrrolidone (NMP) were purchased from Sigma-Aldrich. All reagents were used as received without further purification.

**Fabrication of δ-$MnO_2$ nanostructures.** Powder synthesis included mixing $MnCO_3$ and $K_2CO_3$ powders in a molar ratio of 40 : 9, by milling in isopropyl alcohol for 10 min using a McCrone Micronizing Mill (McCrone Group, USA) with alumina media. The resulting suspension was dried on a hot plate for 30 min at 60 °C and then calcined in alumina crucible at 800 °C for 24 h in air. 0.5 g of the resulting layered $K_xMnO_2$ was proton ion-exchanged in HCl solution (1 mol l$^{-1}$, 45 ml) in an ultrasonic bath at room temperature for 4 h. The ion exchange process was repeated two additional times, followed by washing with DI water and air drying. The absence of K ions was confirmed by energy dispersive spectroscopy and XPS survey scans shown in Supplementary Figs 1 and 2. In addition, the XPS survey scans showed a complete absence of any signal from Al ions, giving good confidence that no contamination by the $Al_2O_3$ milling media was encountered.

To obtain exfoliated $MnO_2$ nanosheets, 0.35 g $H_xMnO_2$ was equilibrated with 32.5 ml aqueous TBAOH solution (30 ml $H_2O$ + 2.5 ml 40 wt% TBAOH) for 4 h in an ultrasonic bath at room temperature. The resulting suspension was centrifuged at 10,000 r.p.m. for 10 min. to separate the remaining bulk $H_xMnO_2$ from the nanosheet suspension. Self-assembly of the nanosheets in the colloidal suspension was achieved by adding 6N HCl solution to the suspension at a constant rate of 1 ml min$^{-1}$, while stirring to reach pH = 2, resulting in flocculation to form high-surface area 3D porous structures.

Mn defects and redox were controlled by equilibrating the assembled nanosheets at target pH values by increasing the pH from 2 upwards using 1 mol l$^{-1}$ NaOH additions and stirring for 24 h. Dry nanosheet assemblies were finally obtained by washing, centrifugation, rinsing in 2-propanol and drying overnight at room temperature. Thus, 3D porous $MnO_2$ nanostructures assembled from ultrathin 2D δ-$MnO_2$ nanosheets with controlled defect content were obtained. For comparison with the 3D assembled nanosheets, protonated bulk $H_xMnO_2$ without exfoliation was also electrochemically tested.

**Characterization of the samples.** Microstructures were studied using scanning electron microscopy (SEM, FEI Quanta 200) at 20 kV. Transmission electron microscopy and high-resolution SEM were carried out using a Hitachi HF-3300 TEM/STEM. The STEM unit has a secondary-electron detector, which allows simultaneous high-resolution SEM and TEM imaging at 300 kV. The thickness and crystallite dimension of the exfoliated $MnO_2$ nanosheets were probed by multimode atomic force microscope (AFM, Bruker Dimension Icon) in tapping mode using antimony doped silicon tips. The exfoliated nanosheets were electro-statically attached to a clean silicon wafer by drying the nanosheet suspension, and imaged in air. Thermogravimetric analysis (TGA) was performed using a TA Instruments SQT-Q600 DTA/TGA under flowing air with a heating rate of 10 °C min$^{-1}$. The presence of any functional groups from organic compounds was probed with infrared spectroscopy (Nicolet 6700 FT-IR Spectrometer, USA). The specific surface area and porosity were examined by nitrogen adsorption and desorption isotherms collected at 77 K using a Micromeritics TriStar II 3020 system. The local chemical environment of the samples was characterized using a PHI Quantera X-ray photoelectron spectrometer equipped with Al Kα radiation. Mn K-edge X-ray absorption near-edge spectroscopy measurements were carried out at the bending magnet beamline F3 at the Cornell High Energy Synchrotron Source (CHESS). Data were collected at room temperature in fluorescence mode using a Hitachi vortex 4-element silicon drift detector. All spectra were calibrated using the spectrum of Mn metal foil, and the software package ATHENA was used for analysis[66]. The Mn vacancy content in δ-$MnO_2$ samples was determined using high-energy X-ray scattering and PDF analysis, with data collected at the Advanced Photon Source on beam-line 11-ID-B using the rapid-acquisition PDF geometry[67]. Data sets were collected using standard 1 mm Kapton capillaries in Debye-Scherrer geometry, Si<311> monochromated primary beam at 58.66 keV, and a silicon flat plate area detector. Scattering data for PDF extraction were collected over a Q-range of 0.4–24.5 Å$^{-1}$, and the powder diffraction data for all samples was collected over a Q-range of 0.2–9.0 Å$^{-1}$. 2D X-ray diffraction data were integrated to 1D using FIT2D[68], after appropriately calibrating detector deviations from orthogonality and masking invalid pixels. $CeO_2$ was used as the calibration standard for detector geometry. Meanwhile, $CeO_2$ and Ni were used to evaluate the instrument response function. The PDF data was reduced using PDFgetX2 (ref. 69), which includes the appropriate corrections for inelastic scattering and energy-dependent detector response, in addition to experimental background and absorption corrections, amongst others.

**Electrochemical measurements.** The working electrode was prepared by mixing 80 wt% active material, 15 wt% acetylene black and 5 wt% PVDF in NMP solution. After stirring for 6 h, the homogeneous slurry was spread onto a Ni foil substrate with an area of 1 cm$^2$, and then heated at 100 °C for 2 h to evaporate the solvent and obtain the electrode. The loading of the active material on the working electrode was in the range of 0.5–0.6 mg cm$^{-2}$. The capacitive performance was measured using a CHI 650E electrochemical analyser (CHI, USA) with a conventional three-electrode cell. Ag/AgCl and platinum wire were used as the reference and auxiliary electrodes, respectively, with 1 M $Na_2SO_4$ aqueous

electrolyte. Cyclic voltammetry scans were carried out from 0 to 1 V at a scan rate of 50 mV s$^{-1}$. Galvanostatic charge–discharge was measured at different constant current densities from 0.2 to 10 A g$^{-1}$. EIS was performed in the frequency range of 0.1 Hz–100 kHz at an open circuit potential of 5 mV. EIS data was fitted to an electrical equivalent circuit model using ZsimpWin (Version 3.21, EChem Software) software.

**Data availability.** The data that support the findings of this study are available from the authors on request.

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

## Acknowledgements

This material is based upon work supported by the National Science Foundation under Grant No. DMR-1409102. This research used resources of the Advanced Photon Source, a U.S. Department of Energy (DOE) Office of Science User Facility operated for the DOE Office of Science by Argonne National Laboratory under Contract No. DE-AC02-06CH11357. We are indebted to K. Chapman at the Advanced Photon Source for valuable guidance in high-energy X-ray measurements. XANES measurements were conducted at the Cornell High Energy Synchrotron Source (CHESS), which is supported by the National Science Foundation and the National Institutes of Health/National Institute of General Medical Sciences under NSF award DMR-1332208.

## Author contributions

S.T.M., D.L. and D.D.E. conceived the research. P.G. and T.H. synthesized the materials. P.G. performed the electrochemical measurements. P.M. and S.T.M. performed the X-ray scattering measurements. P.G. and R.H. performed the XANES measurements. Y.G. performed the XPS and AFM measurements. J.Y.H. performed the TEM measurements. P.G., P.M. and T.H. performed the material characterization. All authors discussed and analysed the data. P.G., P.M. and S.T.M. wrote the paper. All authors discussed and commented on the manuscript.

## Additional information

**Competing financial interests:** The authors declare no competing financial interests.

