## [Peer Review File · Nature Communications]

Reviewers' comments:

Reviewer #1 (Remarks to the Author):

The critical role of point defects in improving the specific capacitance of δ -MnO₂ nanosheets

This paper reports industrially scalable synthesis of δ -MnO₂ via multi step process: a thermal calcination followed by a proton exchange, a exfoliation, and two pH control steps. The optimal material's capacitance is as high as 300 F g⁻¹. This work also observes Mn⁴⁺ → Mn³⁺ reduction and Mn ions migration from the plane to the surface, creating Mn vacancy. The Mn vacancy improves the specific capacitance, charge transfer resistance, and cycling stability. Although the performance in supercapacitor is improved - however, the cycling decay is still problematic.

1. How were MnCo₃ and K₂CO₃ mixed? Mortar and pestle or other milling techniques? What was the condition?
2. How much KxMnO₂ powder is proton exchanged from the 45 ml HCl solution?
3. XPS survey scans should be provided to supplement the EDS result indicating that the potassium fully exchanged.
4. The IR and TGA shows some presence of TBAOH in the sample. This can also be confirmed with XPS survey scans. The authors can discuss the impact of the TBAOH impurity on the electrochemical performance.
5. In the parent KxMnO₂, the synchrotron diffraction data (FigS3) reveals that there is 9% rhombohedral impurity. The diffraction pattern for the parent KxMnO₂ is fitted but the next step materials are not fitted. Can the authors discuss the possible impact of this impurity on the electrochemical performance?
6. Fig.2 clearly present not only the broadening, but also peak shifts between protonated MnO₂ and the reassembled MnO₂s. Furthermore, there are MnO₂ phases such as alpha, beta, gamma, ramsdellite, etc. which can exhibit hk0 peaks. Please elaborate more on how the reassembled MnO₂s are still delta phase.
7. The X-ray scattering and PDF shows the presence of defects and the XANES reveals that the overall oxidation state is different among samples. Are all Mn³⁺ associated with the defect formation or could the Mn³⁺ be independent of the defect. Please add some discussion about the effect of oxidation state on the electrochemical performance.
8. The mass loading is 0.55 mg cm⁻² and the gravimetric capacitance is 300 F g⁻¹, meaning the areal capacitance is 0.17 F cm⁻². Is it possible to increase the mass loading while maintaining the low resistance and high gravimetric capacitance?
9. Fig.6a scheme and the authors' choice of word "surface" or Frenkel defect sounds like the bulk Mn is migrating to outer most surface of the nanosheet particle. However, such surface Mn ion may be prone to dissolution. It may be more likely for the in-plane Mn to migrate towards the adjacent Mn-O slab. If the Mn migrates to the interslab between the two Mn-O layers, this will explain/support the PDF result better.

Conclusion:

Overall, this work reports a high capacitance δ -MnO₂ material. The authors attributes the high capacitance to the presence of Mn vacancies. The material is well characterized to confirm the phase of MnO₂ and the presence of the vacancies are convincing. The authors still need to address

the issue of capacity fading upon cycling. I recommend this manuscript be published in Nature Communication with major revision.

Reviewer #2 (Remarks to the Author):

This paper reports the effects of vacancy defects in MnO₂ nanosheet on its electrochemical performance. The content of surface Mn vacancy was controlled via reassembly in the HCl solution with various pH values and estimated on the basis of the PDF analysis. Then the electrochemical performance of these reassembled nanosheets was compared. The results showed that the cation defects are correlated with the specific capacitance. However, some important points are not well illustrated. And the novelty is not high enough. I would not suggest this work to be published in Nature Communications on the basis of the following points.

1. Exfoliation of MnO₂ nanosheet from its layered bulk materials and then reassembly/flocculation of these nanosheets with introduced cation species to form a 3D porous network are well-developed methods and have been reported by numbers of papers. Moreover, the quality of exfoliated MnO₂ nanosheets in this work is not so good. Most of these MnO₂ nanosheets are incompletely exfoliated nanosheets of >5 layers, which will lead to significant reduction of the active areas for electrochemical reactions. In addition, the size of these nanosheets is decreased to around 150 ~ 200 nm compared to its original layered bulk HxMnO₂ (1 ~ 2 μm), which indicates the serious lateral fracture during the sonication exfoliation process.

2. It is well known that Mn⁴⁺ is reduced to Mn³⁺ to maintain the charge balance during the intercalation of cation species within the birnessite MnO₂ layers. Thus, it is common that the content of Mn³⁺/Mn⁴⁺ can be controlled by adjusting the concentration of intercalated cation species, i.e. the pH of the HCl solution in this work. Besides, the model of surface Mn vacancy/defect has been proposed for K-birnessite MnO₂ as well as birnessite nanosheets. It may be overstatement to claim that this work is the first time to control the Mn point defects and Mn³⁺/4⁺ ratios of MnO₂ nanosheets.

3. Although the authors compare the electrochemical performance of these reassembled nanosheets, a difference between pH=2 and pH=4 samples is not obvious. Control of the defects does not seem an effective method to increase the intercalation capacitance. In addition, in Figure 7(a), there is a pair of broad peaks in CV curve of reassembled MnO₂ at pH 2, while no such peaks were observed for reassembled MnO₂ at pH 4. The different behaviors between these samples should be further explained.

Reviewer #3 (Remarks to the Author):

In this manuscript, Gao et al. prepared 3D porous MnO₂ nanosheets using KxMnO₂ as starting materials and following proton ion-exchange, exfoliation and re-assembly. By controlling the pH value of re-assembly condition, MnO₂ materials with different intentional defects were obtained. The interesting finding is that intentional defects resulted in better electrochemical performance when the as-prepared MnO₂ materials were evaluated as capacitor electrodes. Overall, this study is interesting. However, several issues need to be addressed before acceptance for publication.

1. The authors compared the re-assembled MnO₂ sample with protonated MnO₂ in terms of surface area, electrochemical performance. Protonated MnO₂ is a bulk material. An exfoliation step was conducted with TBAOH solution under ultrasonication. It is better to compare these properties of exfoliated MnO₂ and MnO₂ materials with different intentional defects.

2. In the caption of Figure 1c, it indicated high resolution SEM image of re-assembled MnO₂ treated in pH=2 solution. However, in the main text, it indicated exfoliated MnO₂. Please modify

the sentence or the figure caption.

3. TGA results confirmed the existence of H₂O and residual of TBAOH in reassembled MnO₂. Is there any influence from these H₂O and TBAOH on the electrochemical performance?

4. The authors claimed that "three samples exhibit typical rectangular shapes, which indicates ideal capacitive behavior" and "the linear variation of the potential with time and the symmetric charge-discharge characteristics are indicative of capacitive behavior". In the introduction part, the authors have also mentioned the two types capacitors, which are electrical double-layer capacitors and pseudocapacitors. Clearly, Figure 7a showed some redox peaks rather than rectangular shapes. Please modify the sentence. In the charge-discharge process (Figure 7b), obviously, the curves were not linear. Moreover, the discharge time is much shorter than the charge time, which indicated a limited efficiency. This phenomenon is not typical for capacitors. Please explain.

5. As a capacitor, it is necessary to run a long-term cycling. Only 1000 cycle was provided in this work.

Reviewer #1:

This paper reports industrially scalable synthesis of δ -MnO₂ via multi step process: a thermal calcination followed by a proton exchange, a exfoliation, and two pH control steps. The optimal material's capacitance is as high as 300 F g⁻¹. This work also observes Mn⁴⁺ → Mn³⁺ reduction and Mn ions migration from the plane to the surface, creating Mn vacancy. The Mn vacancy improves the specific capacitance, charge transfer resistance, and cycling stability. Although the performance in supercapacitor is improved – however, the cycling decay is still problematic.

(1) How were MnCO₃ and K₂CO₃ mixed? Mortar and pestle or other milling techniques? What was the condition?

The text has been updated in the experimental procedure section to provide this information. The improved text in §Methods: Fabrication of δ -MnO₂ nanostructures (page 20) of the manuscript reads as follows:

“Powder synthesis included mixing MnCO₃ and K₂CO₃ powders in a molar ratio of 40 : 9, by milling in isopropyl alcohol for 10 minutes using a McCrone Micronizing Mill (McCrone Group, USA) with alumina media. The resulting suspension was dried on a hot-plate for 30 min at 60°C and then calcined in alumina crucible at 800 °C for 24 h in air.”

(2) How much K_xMnO₂ powder is proton exchanged from the 45 mL HCl solution?

The experimental procedure has been updated to provide more details, and the improved text in §Methods: Fabrication of δ -MnO₂ nanostructures (page 20) of the manuscript reads as follows:

“0.5 g of the resulting layered K_xMnO₂ was proton ion-exchanged in HCl solution (1 mol/L, 45 mL) in an ultrasonic bath at room temperature for 4 h. The ion exchange process was repeated two additional times, followed by washing with DI water and air drying.”

(3) XPS survey scans should be provided to supplement the EDS result indicating that the potassium fully exchanged.

We have added the XPS survey scan into the supplemental material (Supplementary Fig. 2). In addition, we refer to this data in the main text, and also comment on the absence of Al

contamination from the milling operation. §*Methods: Fabrication of δ -MnO₂ nanostructures* (p. 21) now reads as follows:

“The absence of K ions was confirmed by energy dispersive spectroscopy and XPS survey scans in Supplementary Figs 1 and 2. In addition, the XPS survey scans showed a complete absence of any signal from Al ions, giving good confidence that no contamination by the Al₂O₃ milling media was encountered.”

(4) The IR and TGA shows some presence of TBAOH in the sample. This can also be confirmed with XPS survey scans. The authors can discuss the impact of the TBAOH impurity on the electrochemical performance.

As noted in response to the previous comment, we have added the XPS survey scan in Supplementary Fig. 9. In agreement with the IR and TGA data, the XPS survey scan demonstrates the presence of TBAOH after the exfoliation and flocculation processes. The XPS spectrum contains a nitrogen 1s peak and a small peak located at around 288 eV that is attributed to carbon bound to nitrogen, and these areas attributed to TBAOH.

With respect to the effect of TBAOH on electrochemical performance, we used IR to show that the TBAOH is removed from the MnO₂ nanosheet assemblies after only one charge-discharge cycle. The new IR data is shown in Supplementary Fig. 10. We further revised the text on pages 7 and 8 of the manuscript to read as follows:

“The presence of residual TBAOH in the reassembled samples was indicated by a small mass loss (~3 %, corresponding to 0.01 TBAOH per MnO₂ formula unit and therefore about 9 % surface coverage) in the temperature region of 200-400 °C^{57, 58}. Infrared (IR) spectroscopy (Supplementary Fig. 8) and X-ray photoelectron spectroscopy (XPS) (Supplementary Fig. 9) were also used to detect remnant TBAOH in the nanosheet assemblies after processing. IR was used to show that the TBAOH is removed from the nanosheet assemblies after one charge-discharge cycle, as shown in Supplementary Fig. 10. Although it is not possible to completely remove the TBAOH molecules from the nanosheet assemblies by washing, its small surface coverage (~9 %) and easy extraction during the first electrochemical cycle together suggest that it has limited if any influence on the measured electrochemical performance.”

(5) In the parent K_xMnO₂, the synchrotron diffraction data (Fig. S3) reveals that there is 9%*

*rhombohedral impurity. The diffraction pattern for the parent $K_x\text{MnO}_2$ is fitted but the next step materials are not fitted. Can the authors discuss the possible impact of this impurity on the electrochemical performance? *Note: this is now Supplementary Figure 5 in the updated version.*

Perhaps we should have been more clear about the nature of the nanosheet. The rhombohedral and monoclinic polytypes are both comprised of $\delta\text{-MnO}_2$ nanosheets, and only the stacking of these sheets is different. Therefore, the rhombohedral birnessite should not be considered an ‘impurity’ because, after exfoliation, the MnO_2 nanosheets are all identical. The paper by Drits, *et al.*¹, for example, describes this and other details of stacking of $\delta\text{-MnO}_2$ nanosheets in detail.

To improve and clarify the manuscript, the text in the supplemental information (page 6) has been modified from “polymorph” to “polytype,” which appropriately reflects the structural relationship of the monoclinic and rhombohedral structures.

1. V. Drits, B. Lanson, and A. Gaillot, “Birnessite polytype systematics and identification by powder X-ray diffraction,” *Am. Miner.* 92(5-6), 771-788 (2007).

(6) Fig. 2 clearly presents not only the broadening, but also peak shifts between protonated MnO_2 and the reassembled MnO_2 s. Furthermore, there are MnO_2 phases such as alpha, beta, gamma, ramsdellite, etc. which can exhibit $hk0$ peaks. Please elaborate more on how the reassembled MnO_2 s are still delta phase.

Even in poorly-crystalline manganates where the X-ray scattering profile may be ambiguous, the derived pair distribution function can identify the polymorph with good certainty. For example, Zhu *et al.*² show a collection of PDFs for MnOOH , pyrolusite, cryptomelane, todorokite, and a collection of birnessite or $\delta\text{-MnO}_2$ type structures. The intensity ratios of the first several correlation peaks clearly distinguish these polytypes clearly.

To improve the manuscript, we have clarified this point further by the addition of Supplementary Fig. 6. A calculated PDF for a single $\delta\text{-MnO}_2$ sheet is compared to the experimental data. While a single sheet is clearly not representative of the data, the relative features of the calculated PDF clearly identify the experimental structure as possessing the $\delta\text{-MnO}_2$ motif.

The peak shifts in Fig. 2, meanwhile, are attributable to the change in interlayer ordering. Increased quantities of water found in the interlayer galleries of reassembled MnO_2 nanosheets shift the basal series ($00l$ peaks) towards larger spacings. Elimination of lateral intersheet order

breaks the symmetry of the parent birnessite and results in diffuse intensity profiles in the *hkl* region. These features are routinely encountered in phyllosulfates and other layered minerals (e.g. phyllosilicates).

To clarify the peak shifts, the text in §*Phases and microstructure* on page 5 has been modified to include the statement:

“Shifts of the basal reflections result from increased water content in the reassembled nanosheet floccules compared to the proton-exchanged form.”

2. M. Zhu et al., “Structure study of biotic and abiotic poorly-crystalline manganese oxides using atomic pair distribution function analysis,” *Geochim. Cosmochim. Acta* 81, 39-55 (2012).

(7) *The X-ray scattering and PDF shows the presence of defects and the XANES reveals that the overall oxidation state is different among samples. Are all Mn³⁺ associated with the defect formation or could the Mn³⁺ be independent of the defect. Please add some discussion about the effect of oxidation state on the electrochemical performance.*

In order to improve the flow and clarity, we have made some modifications to the second and third sections of §*Results and Discussion*. In §*Redox and Defects* on page 9, we refer to Table 2 explicitly, so that the average oxidation states are both listed in the text and referred to in the Table. Later on pages 10-11, we have added some text to clarify that we are teaming the XANES with the PDF and electrochemical measurements, which reads:

“High energy X-ray scattering and pair distribution function analysis were undertaken to further characterize the defects formed in soft chemically reduced MnO₂ nanosheets, with the specific goal of correlating the quantity of Mn point defects with Mn reduction and electrochemical performance.”

At the end of §*Redox and defects* on pages 12-13, we have added another reference to Table 2 that provides the amount of Mn in the 3 and 4+ states as compared to the Mn defect content. The improved text now includes the statement:

“Table 2 shows that approximately 2/3, 1/2, and 1/4 of the Mn ions remaining in the nanosheets are reduced to Mn³⁺ for the pH = 2, 4, and H_xMnO₂ samples, respectively. The PDF and XANES analyses are therefore complementary in

demonstrating that surface Frenkel defects and Mn reduction are more favorable in high surface area nanosheet assemblies, with steric limitations reducing the extent of the reactions in well-crystalline birnessites.”

The above changes then lead into the existing text late in §*Electrochemical measurements* where we refer again to Table 2 and correlate the Mn surface Frenkel defects with Mn oxidation state and Na loading during electrochemical cycling. We believe that these minor changes have improved the flow and coherency of the arguments.

(8) The mass loading is 0.55 mg cm^{-2} and the gravimetric capacitance is 300 F g^{-1} , meaning the areal capacitance is 0.17 F cm^{-2} . Is it possible to increase the mass loading while maintaining the low resistance and high gravimetric capacitance?

This is an excellent question in terms of technology deployment. While the thesis of the manuscript is our ability to quantify and correlate the point defects with performance, our group has begun some optimization work. We have proven that we can get to $\sim 3 \text{ mg}\cdot\text{cm}^{-2}$ while retaining most of the capacitance, and we’re currently exploring the use of electrophoretic deposition of nanosheets onto Ni foam substrates as well as graphene composites to further improve the mass loadings. While we could include some of these more recent results in the supplemental information, we prefer to keep the paper focused on the effects of defects on capacitance, which has not been reported before.

(9) Fig. 6a scheme and the authors’ choice of word “surface” or Frenkel defect sounds like the bulk Mn is migrating to outer most surface of the nanosheet particle. However, such surface Mn ion may be prone to dissolution. It may be more likely for the in-plane Mn to migrate towards the adjacent Mn-O slab. If the Mn migrates to the interslab between the two Mn-O layers, this will explain/support the PDF result better.

We chose to use the term “surface Frenkel” defect to indicate that the Mn^{3+} ion migrates from the plane of the sheet to either side of the sheet. In our nanosheet assemblies, with high surface area, we assume that statistically many of those will be on the exposed surfaces (compared to between sheets as is the dominant mechanism in crystalline birnessites that have not been exfoliated). Our processing approach yields some isolated sheets and some re-stacked sheets, so it is true that some of the surface Mn^{3+} will be between sheets, but a major fraction will be on the surfaces. We are confident from our work (and from the literature, including the work by Manceau, et al. in reference 30 of the manuscript) that there are no pair correlation peaks in the PDF data that suggest Mn^{3+} - Mn^{3+} interactions between sheets. For example, if the Mn^{3+} ions were to link

face-to-face, we would see another clear PDF peak, which is absent in the experimental data.

Conclusions:

Overall, this work reports a high capacitance δ -MnO₂ material. The authors attributes the high capacitance to the presence of Mn vacancies. The material is well characterized to confirm the phase of MnO₂ and the presence of the vacancies are convincing. The authors still need to address the issue of capacity fading upon cycling. I recommend this manuscript be published in Nature Communication with major revision.

The reviewer raises a good point in that we have not uncovered the origin of the cycling fade, a topic which in general remains poorly understood, even for bulk electrodes. We will point out that we used a very high current density, roughly 5 times higher than reported for MnO₂ nanostructures by Song, et al.³ (for example), to show the worst-case scenario. Indeed, Song and coworkers showed only about 92 % capacity retention after 1000 cycles when using current density 5 times smaller than that used in our study.

To improve the manuscript, we have included the likely origins of the capacity fade and highlighted our use of very high current density for the evaluation. §*Electrochemical measurements* on page 16 now reads as follows:

“Our ongoing work involves tracking the reversibility of the electrochemical strains induced during cycling, and although we have not yet uncovered the origins of the cycling fade, our results are promising when compared to earlier work, for example for MnO₂ nanostructures which retain ~92% of the initial capacity after 1000 cycles when using a current density 5 times smaller.”

3. Min-Sun Song et al., “Porously Assembled 2D Nanosheets of Alkali Metal Manganese Oxides with Highly Reversible Pseudocapacitance Behaviors,” *J. Phys. Chem. C* 114, 22134–22140 (2010).

Reviewer #2:

This paper reports the effects of vacancy defects in MnO₂ nanosheet on its electrochemical performance. The content of surface Mn vacancy was controlled via reassembly in the HCl solution with various pH values and estimated on the basis of the PDF analysis. Then the electrochemical performance of these reassembled nanosheets was compared. The results

showed that the cation defects are correlated with the specific capacitance. However, some important points are not well illustrated. And the novelty is not high enough. I would not suggest this work to be published in Nature Communications on the basis of the following points.

(1) Exfoliation of MnO₂ nanosheet from its layered bulk materials and then reassembly/flocculation of these nanosheets with introduced cation species to form a 3D porous network are well-developed methods and have been reported by numbers of papers.

While we agree completely that MnO₂ nanosheet assemblies have been prepared earlier and tested for supercapacitor and other applications, we would like to clarify the novelty of our work is that we have prepared these nanostructures and intentionally controlled the defect content, then quantified the defect content experimentally -- to allow the first direct connection between the defects, Mn redox, and capacitance.

(1b) Moreover, the quality of exfoliated MnO₂ nanosheets in this work is not so good. Most of these MnO₂ nanosheets are incompletely exfoliated nanosheets of >5 layers, which will lead to significant reduction of the active areas for electrochemical reactions. In addition, the size of these nanosheets is decreased to around 150~200 nm compared to its original layered bulk H_xMnO₂ (1~2 μm), which indicates the serious lateral fracture during the sonication exfoliation process.

To the reviewer's point about the quality of the nanosheets, a challenge in this system is quantification of the micro/meso structures using microscopy. A recent paper by Sun, *et al.*⁴ in *Nature Communications* shows that assembly of oxide nanosheets is possible, and their SEM and TEM images show nanostructures of quality similar to those we present in our manuscript. However, we will point out that the supplemental data⁵ for the paper by Sun, *et al.*⁴ shows XRD patterns that are in most cases typical of fairly well-crystalline oxides despite the porous and thin-walled microstructures evident in the SEM photos. This result suggests that these authors encountered either some incomplete exfoliation or extensive restacking after exfoliation. In addition, their AFM images show stacks of 3-5 sheets for MnO₂, TiO₂, ZnO, WO₃ and Co₃O₄ nanosheets, a situation we also report in our supplemental data. Another work by Song and coworkers³ shows XRD patterns similar to those we've obtained, where we estimate the diffracting component contributing to the 00l diffraction lines is roughly 5 nanosheets thick. However, it is critical to note that the TEM shows many single-layered nanosheets and these will not contribute to the 00l peaks in the XRD patterns. Finally, the work by Song *et al.*³ reports specific surface areas for their nanosheet assemblies of only 50 – 70 m²·g⁻¹, compared to our values of 120 and 144 m²·g⁻¹.

Based upon these comparisons (and similar examples from the literature on TiO_x nanosheets that are perhaps not necessary to include here), we would argue that our exfoliation and processing is of equal or better quality than similar work in the literature. We have improved the manuscript by adding reference to the papers by Song, et al. and Sun, et al., such that §Phases and microstructure of page 6 reads as follows (new text in red):

“Nitrogen adsorption-desorption isotherms were used to quantify the specific surface area (SSA) of all specimens as shown in Fig. 3. While protonated MnO_2 showed no evidence of mesopores, the reassembled MnO_2 nanostructures show typical type IV isotherms (IUPAC classification) with distinct H3-type hysteresis loops, a result of open slit-like mesopores⁴⁸. The Brunauer Emmet and Teller (BET) specific surface areas (SSA) were $120 \pm 0.4 \text{ m}^2 \cdot \text{g}^{-1}$ for the pH = 2 sample and $144 \pm 1 \text{ m}^2 \cdot \text{g}^{-1}$ for the pH = 4 sample, and only $4.5 \pm 0.1 \text{ m}^2 \cdot \text{g}^{-1}$ for H_xMnO_2 , where the former values are roughly double the surface areas reported by Song, et al.⁴⁹ for reassembled oxide nanosheets. Together, the XRD, BET and microscopy show that the re-assembled nanosheets have macro and mesopores, with the mesoporosity arising due to loose agglomeration of randomly oriented sheet clusters. The extent of exfoliation and/or restacking, typical sheet thicknesses determined by AFM and average crystallinity in the sheet stacking direction as determined by XRD are similar or better for our MnO_2 specimens than those reported recently for MnO_2 , TiO_2 , Co_3O_4 , ZnO , and WO_3 , for example⁴⁴. Therefore, the typical structures in Fig. 1(d) are of the form of edge-to-face assembled nanosheet booklets, with “wall thicknesses” of up to 4 nm. The high SSA of the MnO_2 nanosheet assemblies facilitates infiltration of the electrolyte into the porous electrode to enhance the specific capacitance^{6, 50}.”

4. Sun et al., “Generalized self-assembly of scalable two-dimensional transition metal oxide nanosheets,” *Nat. Commun.* 5, 3813 (2014).
5. http://www.nature.com/articles/ncomms4813?WT.ec_id=NCOMMS-20140514#supplementary-information

(2) *It is well known that Mn^{4+} is reduced to Mn^{3+} to maintain the charge balance during the intercalation of cation species within the birnessite MnO_2 layers. Thus, it is common that the content of $\text{Mn}^{3+}/\text{Mn}^{4+}$ can be controlled by adjusting the concentration of intercalated cation species, i.e. the pH of the HCl solution in this work. Besides, the model of surface Mn vacancy/defect has been proposed for K-birnessite MnO_2 as well as birnessite nanosheets. It may be overstatement to claim that this work is the first time to control the Mn point defects*

and $Mn^{3+/4+}$ ratios of MnO_2 nanosheets.

We agree with the reviewer on both points about earlier work. Indeed we are not the first to quantify Mn defects in MnO_2 , nor are we the first to control the defect content via pH. However, we are, to the best of our knowledge, the first to correlate the defects with the electrochemical behavior. In addition, we are the first to quantify the defects in exfoliated and reassembled nanosheets. Therefore, the novelty of our work reported in this manuscript is the quantification, on an absolute scale, of the defects and their influence on capacitance and charge transfer resistance.

Furthermore, the nanosheet geometry provides perhaps the most reliable way to make the connection between defects and capacitance, and therefore will be considered highly reliable by the community. In the case of nanoparticles, for example, the initial crystallinity, surface disorder, and surface-to-volume ratios hinder quantitative assessment of surface defects.

(3) Although the authors compare the electrochemical performance of these reassembled nanosheets, a difference between pH = 2 and pH = 4 samples is not obvious. Control of the defects does not seem an effective method to increase the intercalation capacitance. In addition, in Figure 7(a), there is a pair of broad peak in CV curve of reassembled MnO_2 at pH 2, which no such peaks were observed for reassembled MnO_2 at pH 4. The different behaviors between these samples should be further explained.

The reviewer raises a good point about Fig. 7(a) which we have clarified. It is true that we see some intercalation peaks in the CV loop for the pH = 2 sample. We attribute the increasing Mn defect content to improved intercalation, and see these redox peaks in the pH = 3 and 2 samples, but not in the pH = 4 and 9 samples. In the original manuscript, we omitted the data for the pH = 3 and 9 samples, and now we include it in the supplementary data. In addition, we clarify this issue in the manuscript in the first paragraph of §*Electrochemical measurements* on page 15 to read as follows:

“As shown in Fig. 7(a), the CV curves for all three samples exhibit largely rectangular shapes in the potential window from 0 to 1 V, which indicates capacitive behavior. The absence of clear redox peaks for protonated and pH = 4 samples implies that the electrodes are charged and discharged at a pseudo-constant rate over the whole voltammetric cycle. However, the presence of broad redox peaks for the pH = 2 sample indicates that ion intercalation is an active and detectable charge storage mechanism for this specimen. Indeed, even broader redox peaks are noted for

the pH = 3 sample (see Supplemental Fig. 12(a)), while no redox peaks are found for the pH = 4 and pH = 9 (Supplemental Fig. 12(b)) samples. The marked increase in Mn surface Frenkel defects with decreasing pH appears to enhance the intercalation reaction. This is also reflected in Fig. 7(b-d), which show the galvanostatic charge-discharge curves for the samples between 0 and 1 V under different current densities. The potential does not linearly change with time, a behavior typical for pseudocapacitive materials that involve redox reactions.”

Reviewer #3:

In this manuscript, Gao et al. prepared 3D porous MnO₂ nanosheets using K_xMnO₂ as starting materials and following proton ion-exchange, exfoliation and re-assembly. By controlling the pH value of reassembly condition, MnO₂ materials with different intentional defects were obtained. The interesting finding is that intentional defects resulted in better electrochemical performance when the as-prepared MnO₂ materials were evaluated as capacitor electrodes. Overall, this study is interesting. However, several issues need to be addressed before acceptance for publication.

(1) The authors compared the re-assembled MnO₂ sample with protonated MnO₂ in terms of surface area, electrochemical performance. Protonated MnO₂ is a bulk material. An exfoliation step was conducted with TBAOH solution under ultrasonication. It is better to compare these properties of exfoliated MnO₂ and MnO₂ materials with different intentional defects.

Our aim was to focus on the two exfoliated and re-assembled samples with good control of the defects, but we include the protonated (bulk) form for two reasons. Firstly, the quantity of defects created in bulk crystalline birnessite by pH treatment is rather different than in the nanosheets, a point that is new. While the geochemical literature includes exquisitely detailed study of defects in crystalline birnessites, our results show much higher Mn surface Frenkel defect populations allowed by the higher surface areas in our nanosheet assemblies. Secondly, we aimed to convincingly show that our X-ray PDF, XPS, XANES and other studies give accurate results for the bulk phase (by comparison with literature, for example the Manceau, et al. paper in reference 30 of the manuscript) before extending to the nanosheets. For these reasons, we feel that it is important to keep the results for the bulk protonated phase in the paper.

(2) In the caption of Figure 1c, it indicated high resolution SEM image of re-assembled MnO₂

treated in pH = 2 solution. However, in the main text, it indicated exfoliated MnO₂. Please modify the sentence or the figure caption.

We appreciate the identification of our error. We have modified the sentence in the first paragraph in §Phases and microstructure on page 4 in the main text, which can be seen in the revised manuscript.

(3) TGA results confirmed the existence of H₂O and residual TBAOH in reassembled MnO₂. Is there any influence from these H₂O and TBAOH on the electrochemical performance?

This comment is similar to that of comment number 4 of reviewer #1, and we have clarified this point to improve the manuscript. We repeat what we stated earlier for convenience:

With respect to the effect of TBAOH on electrochemical performance, we used IR to show that the TBAOH is removed from the MnO₂ nanosheet assemblies after only one charge-discharge cycle. The new IR data is shown in Supplementary Fig. 10. We further revised the text on pages 7 and 8 of the manuscript to read as follows:

“The presence of residual TBAOH in the reassembled samples was indicated by a small mass loss (~3 %, corresponding to 0.01 TBAOH per MnO₂ formula unit and therefore about 9 % surface coverage) in the temperature region of 200-400 °C^{52,53}. Infrared (IR) spectroscopy (Supplementary Fig. 8) and X-ray photoelectron spectroscopy (XPS) (Supplementary Fig. 9) were also used to detect remnant TBAOH in the nanosheet assemblies after processing. IR was used to show that the TBAOH is removed from the nanosheet assemblies after one charge-discharge cycle, as shown in Supplementary Fig. 10. Although it is not possible to completely remove the TBAOH molecules from the nanosheet assemblies by washing, its small surface coverage (~9 %) and easy extraction during the first electrochemical cycle together suggest that it has limited if any influence on the measured electrochemical performance.”

The influence of H₂O on the electrochemical performance has also been discussed and we revised the text on pages 7 and 8 of the manuscript to read as follows:

“The presence of structural water enables rapid proton or alkali cation transport within the interlayer, which is beneficial for increasing the charge storage properties⁶. The slightly higher water content in the pH = 4 vs. pH = 2 sample is a result of its slightly larger surface

area.”

(4) *The authors claimed that “three samples exhibit typical rectangular shapes, which indicates ideal capacitive behavior” and “the linear variation of the potential with time and the symmetric charge-discharge characteristics are indicative of capacitive behavior”. In the introduction part, the authors have also mentioned the two types capacitors, which are electrical double-layer capacitors and pseudocapacitors. Clearly, Figure 7a showed some redox peaks rather than rectangular shapes. Please modify the sentence. In the charge-discharge process (Figure 7b), obviously, the curves were not linear. Moreover, the discharge time is much shorter than the charge time, which indicated a limited efficient. This phenomenon is not typical for capacitors. Please explain.*

This comment is similar to one of the comments of reviewer #2, and reinforces the need to clarify this point. As noted earlier, we have added data for samples treated at pH = 3 and 9 to clarify. For convenience, we repeat the response to reviewer #2, point 3:

The reviewer raises a good point about Fig. 7(a) which we have clarified. It is true that we see some intercalation peaks in the pH = 2 sample. We attribute the increasing Mn defect content to improved intercalation, and see these redox peaks in the pH = 3 and 2 samples, but not in the pH = 4 and 9 samples. In the original manuscript, we omitted the data for the pH = 3 and 9 samples, and now we include it in the supplementary data. In addition, we clarify this issue in the manuscript in the first paragraph of §*Electrochemical measurements* on page 15 to read as follows:

“As shown in Fig. 7(a), the CV curves for all three samples exhibit largely rectangular shapes in the potential window from 0 to 1 V, which indicates capacitive behavior. The absence of clear redox peaks for protonated and pH = 4 samples implies that the electrodes are charged and discharged at a pseudo-constant rate over the whole voltammetric cycle. However, the presence of broad redox peaks for the pH = 2 sample indicates that ion intercalation is an active and detectable charge storage mechanism for this specimen. Indeed, even broader redox peaks are noted for the pH = 3 sample (see Supplementary Fig. 12(a)), while no redox peaks are found for the pH = 4 and pH = 9 (Supplementary Fig. 12(b)) samples. The marked increase in Mn surface Frenkel defects with decreasing pH appears to enhance the intercalation reaction. This is also reflected in Fig. 7(b-d), which show the galvanostatic charge-discharge curves for the samples between 0 and 1 V under different current densities. The potential does not linearly change with time, a

behavior typical for pseudocapacitive materials that involve redox reactions.”

(5) As a capacitor, it is necessary to run a long-term cycling. Only 1000 cycle was provided in this work.

While we agree that additional cycling (and indeed cycling at lower current density as is common for oxide electrodes) would provide new insight, we chose instead to focus the paper on control and quantification of the defects. As noted earlier in response to the last point of reviewer #1, the cycling fade in our materials is in reasonable agreement with what is typically seen for nanoscale oxide electrodes and it is not unusual to report only 1000 cycles. From a practical point of view, preparing and cycling 3 new samples to 3000 or 5000 cycles at lower current densities (as is typical for oxides), would take many weeks and eventually distract from the novelty of the work which is the fundamentally important role of defects.

As we noted in our closing remarks in the manuscript, we are presently attempting to firmly establish the mechanisms of intercalation into/onto the Mn surface Frenkel defects, and we have experiments underway to evaluate cycling fade from the electrochemical expansion/contraction during cycling. We hope to focus on these topics in an upcoming publication.

REVIEWERS' COMMENTS:

Reviewer #1 (Remarks to the Author):

Authors have addressed most of my comments. I will recommend the paper for publication in Nature Comm.